# Amino Acid Analysis and Cytotoxicity Study of Iraqi *Ocimum basilicum* Plant

**DOI:** 10.3390/molecules30153232

**Published:** 2025-08-01

**Authors:** Omar Hussein Ahmed

**Affiliations:** Department of Pharmacognosy and Medicinal Plants, College of Pharmacy, Tikrit University, Tikrit 34001, Iraq; pharmacognosy88@gmail.com; Tel.: +964-7722604040

**Keywords:** amino acids, cytotoxicity, HRT-18 cell line, leaf extract

## Abstract

Background: This paper deals with the detection of amino acid composition of Iraqi *Ocimum basilicum* (basil) leaves and evaluation of the cytotoxic effects of the plant leaf extract on human colorectal cancer cells. Methods: Leaves of *Ocimum basilicum* were collected from Iraq in November 2024. After drying and powdering, the plant material went through cold methanol extraction. Initial phytochemical screening was conducted to identify the presence of alkaloids, flavonoids, coumarins, and terpenoids. Amino acid analysis was completed by an amino acid analyzer with fluorescence detection. The cytotoxic effect was evaluated via the MTT assay on HRT-18 cell lines. Morphological changes were further tested using dual Propidium Iodide/Acridine Orange assay fluorescent staining. Results: Seventeen amino acids were detected in the plant extract. The extract showed dose-dependent cytotoxic effects on HRT-18 cells, with significant reduction in cell viability at concentrations of more than 25 µg/mL. Morphological alterations of membrane blebbing and cell shrinkage were observed, suggesting apoptotic activity. The IC_50_ value confirmed strong cytotoxic potential. Conclusions: The extract of *Ocimum basilicum* leaf cultivated in Iraq shows a rich amino acid profile and significant cytotoxic activity against colorectal cancer cells that highlights its potential effect as a natural source of anticancer compounds.

## 1. Introduction

Medicinal plants have long been a cornerstone of traditional medicine and modern pharmacology, offering a rich source of biologically active compounds with diverse therapeutic applications. Among these, *Ocimum basilicum* Linn., a member of the Lamiaceae family, is widely cultivated in Asia, the Middle East, and the Mediterranean for culinary, aromatic, and medicinal uses [1]. “Sweet basil” is the popular name of *Ocimum basilicum*, which is used in both Ayurvedic and Unani medicine system [2]. There are more than 150 species of the genus Ocimum. Basil is the main crop for the production of an essential oil that is cultivated commercially in many countries [3]. Its popularity is due to its rich and spicy, mildly peppery flavor with trace amounts of clove and mint and it has been widely used as a food ingredient for flavoring enhancement, in meat products and baked foods [4].

The plant is used for its stomachic, antipyretic and alexipharmic effects. It also has emmenagogue and diuretic properties. An infusion of the plant extract is considered to be anthelminthic, antiemetic, diaphoretic and anti-diarrheic in Annam. Diuretic, anti-dysenteric and aphrodisiac effects have also been attributed to the plant seeds. The plant juice exhibits stimulant, carminative and antibacterial effects; its essential oil has antifungal, antibacterial and insecticidal properties [5]. The plant flowers have diuretic, demulcent and stimulant properties. These flowers are also considered to have anti-spasmodic, carminative and digestive stimulant effects [6].

A high concentration of vitamins, minerals and oils is present in the green plant leaves [7]. Phytochemical screening reports for *O. basilicum* showed the presence of the following: proteins, amino acids, glycoside, mucilage, gums, tannins, triterpenoids steroids, phenolic compound, saponins, sterols, flavones and flavonoids. A total of 29 compounds representing 98.0–99.7% of its oils are identified in this plant, which contribute to its pharmacological potential [8,9]. However, limited studies have explored the composition of basil, particularly in local varieties grown in Iraq. An Iraqi study reported that essential oils of *O. basilicum* could be used for pharmaceutical studies and preservatives in the food industry, and was considered to be the first study of the essential components of a new cultivate of Thai basil in Iraq [10]. Profiling amino acids in medicinal plants like *O. basilicum* can provide insights into their nutritional value and enhance their use in dietary supplements and therapeutic formulations [11].

In recent years, increasing attention has been paid to the anticancer properties of medicinal plants. Plant products are promising alternatives or adjuncts to conventional chemotherapy owing to their potential for high efficacy and lower toxicity [12]. One of the most common malignancies worldwide is colorectal cancer, and the study of plant-derived compounds with selective cytotoxic effects is of critical importance [13]. The HRT-18 cell line that is derived from human colorectal adenocarcinoma can serve as a reliable in vitro model for screening cytotoxicity and evaluating apoptosis-inducing activity [14].

This study was carried out to consider two key features of the Iraqi *Ocimum basilicum* plant: (1) to determine its amino acid profile using a high-performance amino acid analyzer, and (2) to evaluate the cytotoxic effects of the plant on HRT-18 colorectal cancer cells by using the MTT assay and Propidium Iodide/Acridine Orange assay fluorescent staining (AO/PI).

## 2. Results

### 2.1. Amino Acid Detection

The chromatographic analysis clearly (Figure 1) demonstrates multiple peaks corresponding to different amino acids present in the extract. The sharp and well-separated peaks indicate good resolution and efficient separation by the amino acid analyzer. The presence of numerous peaks suggests a diverse amino acid profile, which may contribute to the biological activity of the plant. Table 1 illustrates the specific retention times for the 17 amino acids identified in the sample. These times were consistent with standard references, confirming the identity of the compounds. The reproducibility of retention times across multiple runs confirms the reliability and sensitivity of the analytical method. In summary, *Ocimum basilicum* leaves contained 17 amino acids detected by amino acid analyzer.

### 2.2. Cytotoxic Effects Evaluation

Figure 2A shows HRT-18 cells treated with *Ocimum basilicum* extract with noticeable morphological changes observed, such as cell shrinkage, membrane blebbing, detachment from the surface and reduced overall cell density. These features are typical signs of cytotoxicity and apoptosis. In contrast, Figure 2B shows untreated cells that appear well-spread, with normal polygonal shape, forming a dense and uniform monolayer and with no signs of damage. These visual differences support the conclusion that the basil extract induces cell death in cancer cells.

Figure 3 indicates the inhibitory curve, which shows the cytotoxic effect of different concentrations of *Ocimum basilicum* extract on HRT-18 cancer cells. As the concentration of the extract increases, there is a clear increase in the percentage of cell death, indicating a dose-dependent inhibitory effect. Higher doses of the extract can cause greater cytotoxicity, suggesting that the extract contains active compounds capable of killing cancer cells.

Figure 4 represents the dose–response curve used to determine the IC50 value of *Ocimum basilicum* on cancer cells. The *x*-axis represents the concentration of the test compound (in µg), and the *y*-axis represents the percentage of cell inhibition. The IC50 value is represented by the Log of drug concentration of the test for *Ocimum basilicum,* which inhibits 50% of cell viability.

As shown in Figure 5 the post hoc Tukey’s test demonstrates a significant, dose-dependent cytotoxic effect of *Ocimum basilicum* extract on HRT-18 cells. Higher concentrations (particularly 100 µg/mL) show statistically significant reductions in cell viability compared to lower concentrations and the control group (*p* < 0.0001). Minimal or non-significant differences at lower doses suggest that a threshold concentration is needed to elicit a strong cytotoxic response. These results support the extract’s potential as an anticancer agent at higher doses.

### 2.3. DNA Damage Analysis

Figure 6 illustrates a clear difference in fluorescence intensity between treated and control HRT-18 cells, indicating changes in cell viability. In the treated group, fluorescence from dead cells is significantly higher than that from viable cells (*p* < 0.01), while the control group shows a predominance of viable cells with no significant difference between viability and death signals. This suggests that treatment with *Ocimum basilicum* extract induces cytotoxicity in HRT-18 cells, leading to increased cell death.

Figure 7 compares HRT-18 colorectal cancer cells before and after treatment with *Ocimum basilicum* extract. In Figure 7A, the untreated (control) cells show normal morphology, with a flat, polygonal shape, intact membranes, and a dense, healthy monolayer covering the surface. On the other hand, in Figure 7B, the treated cells display clear signs of cytotoxicity, including Cell shrinkage, membrane blebbing, loss of adherence, reduced cell density and rounded or fragmented cells, indicating possible apoptosis. These visual differences support the extract’s antiproliferative and cytotoxic effects on cancer cells.

## 3. Discussion

The amino acid analysis of *Ocimum basilicum* leaves reveals a rich and diverse profile of both essential and non-essential amino acids. A total of 17 amino acids were identified, each with varying concentrations, indicating the nutritional and potentially therapeutic value of this plant. It is important to clarify that the reported amino acid concentrations (e.g., 30 g/100 g) refer to the composition within the methanolic extract, not the whole dry leaf material. The values are expressed per 100 g of crude extract, which may result in higher apparent concentrations compared to values calculated on a whole plant basis. Among the detected amino acids, Phenylalanine exhibited the highest concentration (38.00 g/100 g), followed by Alanine (29.08 g/100 g), Histidine (28.90 g/100 g), and Serine (27.88 g/100 g). The high levels of these amino acids are significant. Phenylalanine is an essential amino acid that is involved in protein biosynthesis and as a neurotransmitter precursor, like dopamine and norepinephrine [15]. Alanine exhibits a vital role in glucose metabolism and production of energy, especially under stress conditions [16]. Other essential amino acids, like Valine (35.65 g/100 g), Leucine (30.65 g/100 g), Isoleucine (23.54 g/100 g), and Methionine (19.08 g/100 g), were also present in considerable amounts. These branched-chain amino acids are principally important for immune function, muscle repair, and neurotransmitter synthesis, making *Ocimum basilicum* potentially beneficial for physical endurance and metabolic health. Interestingly, the high concentration of phenylalanine (38.00 g/100 g) detected in the basil extract may contribute not only to its nutritional value but also to its cytotoxic effects. Phenylalanine has been shown to exert antiproliferative activity through its ability to modulate neurotransmitter pathways and oxidative stress responses in cancer cells. It may also serve as a precursor for secondary metabolites with cytotoxic properties, including phenylpropanoids and flavonoids. Additionally, the presence of other amino acids such as methionine and cysteine, which are involved in redox regulation and glutathione metabolism, may synergistically enhance the apoptotic and oxidative stress-inducing effects observed in HRT-18 cells. These findings suggest that the amino acid composition may play a contributory role in the overall cytotoxic profile of *Ocimum basilicum* extract [1].

The presence of Aspartic acid (17.45 g/100 g) and Glutamic acid (27.44 g/100 g) further supports the plant’s nutritional value. Glutamic acid is a key neurotransmitter and has a role in memory and learning, while aspartic acid is involved in the citric acid cycle and DNA synthesis [17]. Interestingly, Arginine (26.14 g/100 g) and Cysteine (24.15 g/100 g) were also detected in the plant extract. Cysteine contributes to antioxidant defense through glutathione synthesis [18], while arginine supports nitric oxide production, aiding in cardiovascular health and immune modulation [19]. The broad spectrum and high amino acid concentration detected in this analysis support the traditional uses of *Ocimum basilicum* in herbal medicine. The obtained results are consistent with other studies that highlight the nutritional and pharmacological properties of basil species (*Ocimum sanctum* and *Ocimum gratissimum*) for their amino acid content [20,21].

The cytotoxicity assessment of *Ocimum basilicum* extract on HRT-18 colorectal cancer cells reveals significant dose-dependent antiproliferative activity. Figure 2A demonstrates clear morphological changes in treated cells, including shrinkage, membrane blebbing, and detachment, all of which are hallmark indicators of apoptosis and cell stress. In contrast, Figure 2B shows the untreated control group maintaining normal morphology, indicating that the observed effects are indeed induced by the basil extract. Figure 3 illustrates a dose-dependent inhibitory curve, confirming that increasing concentrations of *O. basilicum* extract result in elevated cytotoxic effects. This supports the hypothesis that the plant contains bioactive compounds—such as flavonoids, terpenoids, and phenolics—that interfere with cancer cell metabolism or induce programmed cell death, as supported by literature indicating the anticancer potential of plant-derived compounds [22,23].

The IC50 curve in Figure 4 further quantifies the extract’s potency, showing that the concentration required to inhibit 50% of the cell population is within a practical therapeutic range. This reinforces the potential use of *O. basilicum* in cancer treatment formulations, especially when standardized for high-yielding cytotoxic compounds. The post hoc Tukey’s test shown in Figure 5 provides strong statistical evidence for this cytotoxic effect, particularly at 100 µg/mL, which showed highly significant reductions in cell viability (*p* < 0.0001). The lack of significant difference at lower doses (e.g., 3.125 µg/mL) suggests that a threshold concentration is needed to initiate noticeable cellular damage. These results align with findings from previous studies, which reported similar dose-dependent cytotoxicity of *Ocimum* species on various cancer cell lines [24,25]. Furthermore, the preliminary phytochemical screening conducted in this study confirmed the presence of flavonoids, terpenoids, alkaloids, and coumarins in the methanolic extract of *Ocimum basilicum*. These phytochemicals have been widely reported to possess antiproliferative and pro-apoptotic effects on various cancer cell lines [21,25]. Flavonoids, in particular, are known to induce apoptosis through modulation of oxidative stress and cell cycle arrest, while terpenoids contribute to membrane disruption and mitochondrial dysfunction in cancer cells. Although individual compounds were not isolated in the current study, the cytotoxic effects observed may be attributed, at least in part, to the synergistic activity of these bioactive constituents [25].

The assessment of DNA damage in HRT-18 colorectal cancer cells following treatment with *Ocimum basilicum* extract reveals significant cytotoxic and antiproliferative effects. As illustrated in Figure 6, fluorescence intensity analysis clearly distinguishes between treated and untreated cells. The treated group exhibits markedly increased fluorescence associated with cell death, while the control group maintains high levels of viable cell fluorescence with minimal death signals. The significant difference in viability (*p* < 0.01) confirms that the basil extract induces cytotoxicity, likely through mechanisms involving DNA damage and apoptosis. Figure 7 provides visual confirmation of this effect. Untreated cells (Figure 7A) maintain typical epithelial morphology—polygonal, well-adhered, and forming a confluent monolayer. In contrast, treated cells (Figure 7B) display multiple morphological signs of cytotoxicity, including cell shrinkage, membrane blebbing, and detachment from the culture surface. These features are consistent with apoptotic cell death, supporting the fluorescence data and previous cytotoxicity assays. Although AO/PI staining provides useful insights into the cytotoxic and apoptotic effects of *Ocimum basilicum*, it does not directly quantify DNA strand breaks. Future studies will utilize more sensitive methods, such as the comet assay or TUNEL assay, to accurately evaluate DNA damage at the molecular level.

These findings align with earlier reports that plant-derived compounds, including flavonoids and essential oils found in *O. basilicum*, can cause oxidative stress and DNA fragmentation in cancer cells, leading to apoptosis [26,27]. The observed cell damage and morphological alterations suggest that the extract’s active constituents may disrupt cellular homeostasis and promote DNA damage pathways [28].

One limitation of this study was the lack of scale bars and limited image resolution in microscopy figures, due to equipment constraints. Future studies will aim to include standardized, high-resolution imaging with calibrated scale bars.

## 4. Materials and Methods

### 4.1. Plant Material

*Ocimum basilicum* leaves were collected from a cultivated garden in Tikrit, Iraq (Latitude: 34.615° N, Longitude: 43.678° E) in November 2024. The plant was identified and authenticated by a taxonomist in the Department of Pharmacognosy and Medicinal Plants, College of Pharmacy, Tikrit University. The plants were grown in clay–loamy soil under full sunlight conditions with minimal irrigation and average day temperatures ranging between 20 and 25 °C. Leaves were washed thoroughly, dried under shade, and ground in a mechanical grinder to a fine powder. 

### 4.2. Experimental Work

#### 4.2.1. Extraction Method (Cold Method)

One hundred grams of the powdered plant material was soaked in 1500 mL methanol (HPLC grade, Sigma-Aldrich, St. Louis, MO, USA), with occasional shaking, at room temperature. After 3 days, the methanol-soluble materials were filtered off. The filtrate was evaporated to dryness under a vacuum using a rotary evaporator (Henan Lanphan Industry Co., Ltd., Zhengzhou, China). A dark greenish residue was obtained. The residue evaporated to dryness and was taken to the laboratory for detection of amino acid. The extraction resulted in 5.2 g of dry methanolic extract from the original 100 g of powdered plant material, yielding a percentage extract yield of 5.2%. This dried residue was used for further phytochemical analysis and amino acid profiling. For cytotoxicity studies, the extract was dissolved in DMSO (analytical grade, Sigma-Aldrich, St. Louis, MO, USA) to prepare a stock solution of 100 mg/mL, which was serially diluted to obtain final concentrations ranging from 1000 to 3.125 µg/mL.

#### 4.2.2. Preliminary Phytochemical Examination of Crude Extracts

Phytochemical analysis for the screening and identification of bioactive chemical constituents in the medicinal plants under study was carried out on crude extracts and fractions, as well as powder specimens, using the standard procedures as described [29,30].

Alkaloid test: approximately 0.5 to 0.6 g of each of plant extract and fractions were mixed in 8 mL of 1% HCl (Sigma-Aldrich, St. Louis, MO, USA), warmed, and filtered. Two ml of the filtrate was treated separately with both reagents (Mayer’s and Dragendorff‘s, Sigma-Aldrich, St. Louis, MO, USA), after which it was observed whether the alkaloids were present or absent in the turbidity or precipitate formation.Coumarins test: 0.5 g of each of plant extract and fractions were mixed in a test tube. The mouth of the tube was covered with filter paper treated with 1 N NaOH (Sigma-Aldrich, St. Louis, MO, USA) solution. The test tube was placed for a few minutes in boiling water, and then the filter paper was removed and examined under the UV light detector (Thermo Fisher Scientific, Waltham, MA, USA) for yellow fluorescence indicating the presence of coumarins.Terpenoids test (Salkowski test): 5 mL of each of plant extract and fractions were mixed in 2 mL of chloroform (Sigma-Aldrich, St. Louis, MO, USA) followed by the careful addition of 3 mL concentrated (H_2_SO_4_) (Sigma-Aldrich, St. Louis, MO, USA). A layer of reddish-brown coloration was formed at the interface, thus indicating a positive result for the presence of terpenoids.Flavonoids test: A total of 0.5 g of each of plant extract and fractions were shaken with petroleum ether (Sigma-Aldrich, St. Louis, MO, USA) to remove the fatty materials (lipid layer). The defatted residue was dissolved in 20 mL of 80% ethanol and filtered. The filtrate was used for the following tests:
(a)Three mL of the filtrate was mixed with 4 mL of 1% aluminum chloride (Sigma-Aldrich, St. Louis, MO, USA) in methanol in a test tube, and the color was observed. The formation of a yellow color indicated the presence of flavonoids.(b)Three mL of the filtrate was mixed with 4 mL of 1% potassium hydroxide (Sigma-Aldrich, St. Louis, MO, USA) in a test tube, and the color was observed. A dark yellow color indicated the presence of flavonoids.

### 4.3. Amino Acid Determination

One ml was taken from the extracted sample, 200 µL of ortho-phthalein aldehyde (5%) (Sigma-Aldrich, St. Louis, MO, USA) was added to it and the sample was shaken for two minutes, after which 100 microliters of the last mixture were taken and injected into the amino acid analyzer [31].

The test was conducted in the laboratories of the Scientific Research Authority/Environment and Water Research Center using the amino acid analyzer (L8900, Hitachi High-Technologies, Tokyo, Japan). The method used involved the carrier phase consisting of (methanol: acetonitrile: 5% formic acid) (all from Sigma-Aldrich, St. Louis, MO, USA) was used in proportions (20:60:20) at a flow rate of (1 mL/min). A separation column (C18—NH2 (250 mm × 4.6 mm) was used to separate the amino acids, while a fluorescence detector was used to detect the amino acids at wavelengths (Ex = 445 nm, Em = 465 nm). The (clarity 2015) software version 5.0.5 (DataApex, Prague, Czech Republic) was used to analyze the amino acids [32]. Individual stock solutions (1 mg/mL) were prepared in 0.1 N HCl, and working standards were diluted to prepare calibration curves for each amino acid in the range of 1–100 µg/mL. Quantification was based on comparison of retention times and peak areas of the samples with those of the standards.

### 4.4. Cytotoxicity Assay

#### 4.4.1. Cell Lines Used

HCT-8 [HRT-18] cells were isolated from the large intestine of a 67-year-old, male, adenocarcinoma patient. HCT-8 [HRT-18] is used for cancer and toxicology research.

#### 4.4.2. MTT Cytotoxicity Assay

##### Cell Culture Conditions

The cell lines were cultured in MEM (US Biological, Salem, MA, USA) supplemented with 10% (*v*/*v*) fetal bovine serum (FBS) (Capricorn-Scientific, Ebsdorfergrund, Germany), and 100 IU penicillin, and 100 µg streptomycin (Capricorn-Scientific, Ebsdorfergrund, Germany) and incubated in a humidified atmosphere at 37 °C. Exponentially growing cells were used for the experiments [33].

##### MTT Procedure

Cells were seeded at a density of 10,000 cells in a 96-well microplate (NEST Biotech, (Wuxi, China) and incubated at 37 °C for 72 h until monolayer confluence was achieved. Cytotoxicity was investigated through 3-(4,5-dimethylthiazol-2-yl)-2,5-diphenyltetrazolium bromide (MTT) assay kits that were supplied by Elabscience Biotechnology Co., Ltd. (Wuhan, China). The cells were exposed to a range of concentrations (3.125, 6.25, 12.5, 25, 50, and 100 µg/mL). After 72 h of infection, 28 µL of MTT dye solution (2 mg/mL) was added to each well. The incubation continued for three hours. A total of 100 μL of DMSO was added to each well and incubated for 15 min. The optical density was measured at 492 nm using a microplate reader [33]. Cytotoxicity % was calculated using the following equation:Cytotoxicity % = (OD Control − OD sample)/OD Control × 100,
where OD control is the mean optical density of untreated wells, and OD Sample is the optical density of treated wells [34].

### 4.5. DNA Damage

#### Apoptosis Estimation (Propidium Iodide/Acridine Orange Assay)

The apoptotic attention in cell lines (infected and control) were measured using (AO/PI) from Sigma-Aldrich (St. Louis, MO, USA). A total of 5000 cells/well were seeded in plate, then infected with (*Ocimum basilicum*) for 24 h in a 37 °C incubator. The tested wells received exactly 50 µL of the AO/PI stain mixture (at room temperature) for 30 s. Then, the stain was removed. The images were taken using a Leica fluorescent microscope (Leica Microsystems GmbH, Wetzler, Germany) [35]. Fluorescent intensity was measured by fluorescent microscopy and via image J software (version 1.53t, National Institutes of Health, Bethesda, MD, USA).

### 4.6. Statistical Analysis

The obtained data were statically analyzed using an unpaired *t*-test and Tukey’s ANOVA multiple comparisons test with GraphPad Prism 8 (GraphPad Software, San Diego, CA, USA). The values were presented as the mean ± SD of triplicate measurements [34].

## 5. Conclusions

*Ocimum basilicum* leaf extract exhibits a rich amino acid profile and significant cytotoxic activity against HRT-18 colorectal cancer cells. The extract induced dose-dependent morphological changes, reduced cell viability, and caused DNA damage indicative of apoptosis. These findings highlight its potential as a natural source of anticancer compounds and warrant further investigation into its active constituents and mechanisms of action.

## Figures and Tables

**Figure 1 molecules-30-03232-f001:**
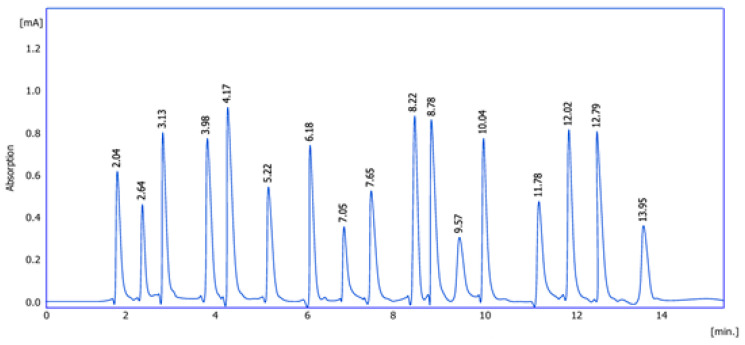
Chromatogram of *Ocimum basilicum* leaves extract.

**Figure 2 molecules-30-03232-f002:**
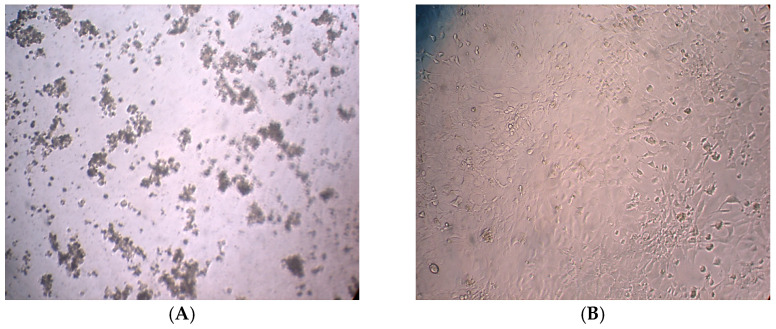
Effect of *Ocimum basilicum* Leaves on HRT-18 Cells. (**A**) Treated HRT-18 Cells. (**B**) Untreated HRT-18. Note: Images were captured using a Leica DM3000 microscope (Leica Microsystems GmbH, Wetzler, Germany) without built-in scale calibration. Scale bars are not available due to equipment limitations at the time of capture.

**Figure 3 molecules-30-03232-f003:**
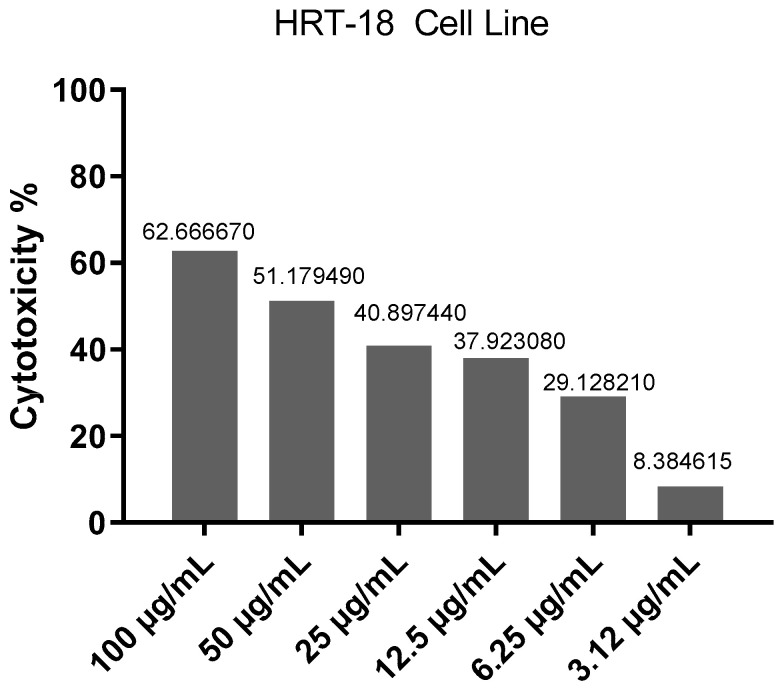
Inhibitory curve showing the effect of *Ocimum basilicum* on cell viability. The *x*-axis represents the concentration of the extract (µg/mL), while the *y*-axis shows the percentage of cell death (cytotoxicity).

**Figure 4 molecules-30-03232-f004:**
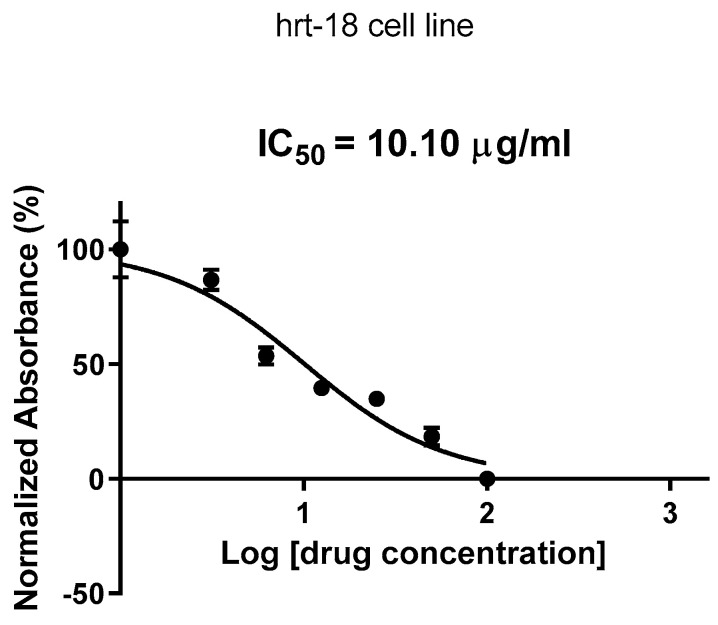
IC50 showing the effect of *Ocimum basilicum* on cell viability. The *x*-axis represents the concentration of the extract (µg/mL), and the *y*-axis shows the percentage of cell inhibition. The IC_50_ value is expressed in µg/mL.

**Figure 5 molecules-30-03232-f005:**
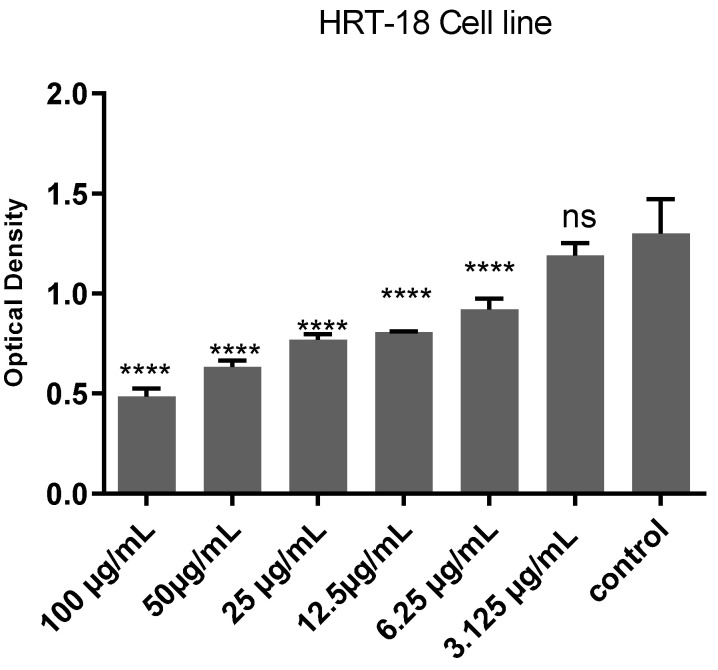
Optical density (OD) reading curve over time for cells treated with *Ocimum basilicum* extract. The *x*-axis represents concentrations in μg/mL, and the *y*-axis represents the OD value at 492 nm. Each data point represents the mean ± standard deviation of triplicates. **** indicates *p* < 0.0001; ns = not significant (*p* > 0.05).

**Figure 6 molecules-30-03232-f006:**
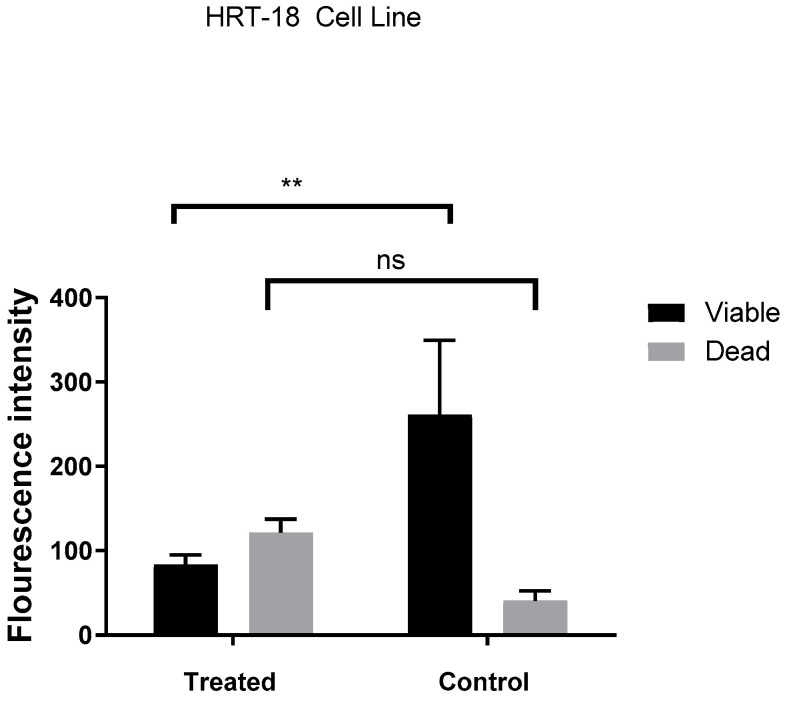
Effect of *Ocimum basilicum* extract on viability and death of HRT-18 cells assessed by fluorescence intensity. ** indicates *p* < 0.01; ns = not significant (*p* > 0.05).

**Figure 7 molecules-30-03232-f007:**
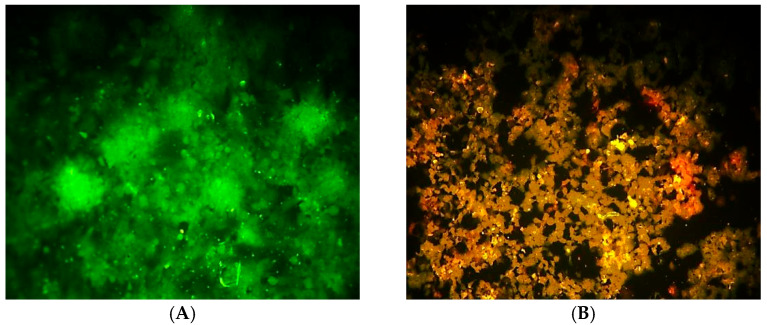
Effect of *Ocimum basilicum* Leaves on HRT-18 Cells. (**A**) Untreated HRT-18 Cells (control). (**B**) Treated HRT-18. Note: Images were captured using a Leica DM3000 microscope without built-in scale calibration. Scale bars are not available due to equipment limitations at the time of capture.

**Table 1 molecules-30-03232-t001:** Retention times of amino acid detected in *Ocimum basilicum* Leaves.

No	Reten. Time [min]	Area [mAU·s]	Height [mAU]	Amount (g/100 g)	Calculation	Peak Type	Compound Name
1	2.04	4562.9	604.7	17.45	Calibration curve	Order	Aspartic acid
2	2.64	4152.6	643.6	26.09	Calibration curve	Order	Glycine
3	3.13	6325.9	794.9	24.12	Calibration curve	Order	Lysine
4	3.98	5854.0	789.8	27.88	Calibration curve	Order	Serine
5	4.17	6214.5	835.7	23.65	Calibration curve	Order	Threonine
6	5.22	5062.6	594.4	23.54	Calibration curve	Order	Isoleucine
7	6.18	6541.8	781.1	29.08	Calibration curve	Order	Alanine
8	7.05	4562.6	386.5	35.65	Calibration curve	Order	Valine
9	7.65	4369.0	549.6	18.99	Calibration curve	Order	Tyrosine
10	8.22	7125.8	827.4	26.14	Calibration curve	Order	Arginine
11	8.78	10,325.6	812.7	24.15	Calibration curve	Order	Cysteine
12	9.57	6521.4	284.8	19.08	Calibration curve	Order	Methionine
13	10.04	10,568.9	741.9	21.65	Calibration curve	Order	Proline
14	11.78	8542.6	418.5	28.9	Calibration curve	Order	Histidine
15	12.02	6985.8	719.4	30.65	Calibration curve	Order	Lucien
16	12.79	11,256.6	712.1	27.44	Calibration curve	Order	Glutamic acid
17	13.95	3565.0	362.3	38.0	Calibration curve	Order	Phenylalanine

## Data Availability

Data will be available upon request.

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
