# Peer review of "Amino Acid Analysis and Cytotoxicity Study of Iraqi Ocimum basilicum Plant"

_molecules, 2025, doi:10.3390/molecules30153232_

Round 1
Reviewer 1 Report
Comments and Suggestions for Authors
In their manuscript "Amino Acid Analysis and Cytotoxicity Study of Iraqi Ocimum basilicum Plant" in this study the author investigate the cytotoxic profile of Iraqi Ocimum basilicum Plant on HRT-18 colorectal-cancer line.
Here are my concerns and questions:
- Why the author only used one type of colorectal cell line and did not include non cancerous normal cell line in this study?
- I believe 30 gram/100 gram of amino acids concentration in plant is very high can you please explain?
- Reference 14 is irrelevant citation.
- If possible please provide a more detail information of the plant used in this study. for example taxonomic authentication and perhaps the condition that this plant was grown in.
- Regarding amino acid analysis detail of reference compounds are missing.
- The microscopic images lack scale bars and are low-resolution please change them to better ones or justify the reason.
- For DNA damage analysis usually comet assay is performed I would recommend the author to measure the DNA damage by comet assay or similar tests.
- In the extraction protocol, extract yield, mass of residue and final concentrations are not mentioned.
Reviewer 2 Report
Comments and Suggestions for Authors
This study examines Iraqi Ocimum basilicum leaf extract, identifying 17 amino acids and demonstrating dose-dependent cytotoxicity on HRT-18 colorectal cells via MTT and AO/PI assays. Methods are sound, results clear, and conclusions aligned with findings. Minor issues include typos and limited focus on Iraqi basil-specific studies. It has moderate originality and significance, suitable for publication with revisions. The following comments and suggestions may help improve the quality of the manuscript:
- Correct spelling errors: 1. “Flourescence” should be changed to ‘Fluorescence’; 2. “Calibration carve” should be changed to “Calibration curve”.
- Are there too many significant digits in the “Cytotoxicity” values in Figure 3?
- It is suggested to add references on Iraqi basilicum to enhance context.
- It is suggested to discuss potential links between specific amino acids (e.g., phenylalanine) and cytotoxicity.
- It is suggested to identify active constituents in the extract (e.g., flavonoids/terpenoids) to strengthen mechanistic insights.
- The units for IC50 in Figure 4 should be clearly stated.
Round 2
Reviewer 1 Report
Comments and Suggestions for Authors
Dear Author,
Thank you for answering all me questions in detail and improve the manuscript.
I can confirm that the manuscript has improved significantly, however a minor issue is that the figure 5 which I believe is the result of the MTT assay the concentrations does not match to the materials and methods and it needs to be adjusted.
I would recommend the author to make it more clear which result belongs to which methods.
Author Response
Reviewer Comment:
A minor issue is that the Figure 5, which I believe is the result of the MTT assay, shows concentrations that do not match those mentioned in the Materials and Methods section. It needs to be adjusted. I would recommend the author to make it more clear which result belongs to which methods.
Author Response:
Thank you for this important observation. As suggested, we have revised the Materials and Methods section (4.4.2.2 – MTT procedure) to match the concentration values used in Figure 5. The updated concentrations are now correctly listed as: 3.125, 6.25, 12.5, 25, 50, and 100 µg/mL. These match the data used to generate the dose-response curve and ICâ‚…â‚€ calculation in the MTT assay.
We appreciate the reviewer’s attention to this inconsistency, which has now been corrected to ensure clarity and consistency in the manuscript.